# Outpatient Readmission in Rheumatology: A Machine Learning Predictive Model of Patient’s Return to the Clinic

**DOI:** 10.3390/jcm8081156

**Published:** 2019-08-02

**Authors:** Alfredo Madrid-García, Judit Font-Urgelles, Mario Vega-Barbas, Leticia León-Mateos, Dalifer Dayanira Freites, Cristina Jesus Lajas, Esperanza Pato, Juan Angel Jover, Benjamín Fernández-Gutiérrez, Lydia Abásolo-Alcazar, Luis Rodríguez-Rodríguez

**Affiliations:** 1Rheumatology Department, and Health Research Institute (IdISSC), Hospital Clínico San Carlos, 28040 Madrid, Spain; 2Dpto. Ingeniería Sistemas Telemáticos, ETSI Telecomunicación, Universidad Politécnica de Madrid, Avda. Complutense 3, 28040 Madrid, Spain

**Keywords:** musculoskeletal diseases, outpatient readmission, predictive model, random forest, quality of life

## Abstract

Our objective is to develop and validate a predictive model based on the random forest algorithm to estimate the readmission risk to an outpatient rheumatology clinic after discharge. We included patients from the Hospital Clínico San Carlos rheumatology outpatient clinic, from 1 April 2007 to 30 November 2016, and followed-up until 30 November 2017. Only readmissions between 2 and 12 months after the discharge were analyzed. Discharge episodes were chronologically split into training, validation, and test datasets. Clinical and demographic variables (diagnoses, treatments, quality of life (QoL), and comorbidities) were used as predictors. Models were developed in the training dataset, using a grid search approach, and performance was compared using the area under the receiver operating characteristic curve (AUC-ROC). A total of 18,662 discharge episodes were analyzed, out of which 2528 (13.5%) were followed by outpatient readmissions. Overall, 38,059 models were developed. AUC-ROC, sensitivity, and specificity of the reduced final model were 0.653, 0.385, and 0.794, respectively. The most important variables were related to follow-up duration, being prescribed with disease-modifying anti-rheumatic drugs and corticosteroids, being diagnosed with chronic polyarthritis, occupation, and QoL. We have developed a predictive model for outpatient readmission in a rheumatology setting. Identification of patients with higher risk can optimize the allocation of healthcare resources.

## 1. Introduction

Readmission can be defined as the return of a patient to a healthcare setting after a discharge. Attention has been mainly focused on readmission following inpatient hospitalization in intensive care, general medicine, and surgery units. In those settings, readmission impose an overwhelming burden at multiple levels [1,2,3,4,5,6,7,8], and has been identified as a major driver of patient’s poor outcomes, such as increased hospital mortality [2,8], length of stay [9], and healthcare costs [10,11]. The percentage of patients undergoing hospital readmission in 30 days following discharge has been reported between 10% and 20% [9,12,13], but this rate increases to 28%–35% when the analysis focuses within 90 days post-discharge [14,15,16].

It is important to point out that readmission is a poorly understood and complex outcome to model. Several factors at different levels, including patient (such as comorbidities, demographic, clinical-related variables, and social determinants of health), center (supply of hospital beds, quality of inpatient care [17]), and health-care system (such as post-discharge follow-up, coordination with primary care) [18,19] interact to affect this outcome. Many tools for inpatient readmission prediction have been developed, with a large heterogeneity regarding the model’s performance [20,21,22].

In the outpatient setting, readmissions have been much less studied. This issue can also be associated with an important burden, affecting particularly those specialties delivering care in outpatient settings, such as rheumatology [23,24]. In those specialties, readmissions can impair the patient’s continuity of care, if those patients returning are not attended by the same physician, and increase the resources consumption, as they will use appointment slots intended for new, previously unattended, patients.

Our ultimate goal is the targeted delivery of inventions aimed at preventing readmission of those patients at greater risk. In order to carry out this objective, first we need to assess the individual patient’s risk to identify candidates for these interventions. Therefore, in the present study our objective is to develop and validate a prediction model of outpatient readmission using the machine learning classifier random forest (RF) [25].

## 2. Patients and Methods

### 2.1. Patients

The Hospital Clínico San Carlos musculoskeletal cohort (HCSC-MSKC) is a routine clinical practice cohort that includes subjects seen at the rheumatology outpatient clinic of our center whose clinical information and management was carried out using a departmental electronic health record (EHR; Medi-LOG), implemented in October 2006. For the present study, data extraction from Medi-LOG was carried out at the end of 2017, and during the following year, data cleaning, quality control, and analysis were performed. Patients were included in this cohort based on data completeness and consistency (see Appendix A “Database Clean-up” and Appendix A for a detailed description). Briefly, patients were included if the first contact in our clinic took place after the implementation of the EHR, and therefore the information from all visits was collected using this tool. In addition, the information stored regarding diagnosis, treatments, and quality of life (QoL) had to be “coherent,” meaning that, for each particular visit, it had to be the same across different tables. After data clean-up, more than 35,000 patients, attending from 1 April 2007 until 30 November 2017 were included in the HCSC-MSKC, with information stored from more than 117,000 visits.

In every patient’s visits, information is collected both as free text and codified. Regarding the former, it includes the clinical notes and comorbidity, or medications prescribed by primary care or other specialists. Regarding the latter, it includes information regarding rheumatology diagnoses (using the ICD9/ICD10), prescribed drugs (using the Spanish Drug and Medical Device Agency (AEMPS) codification system), QoL (using the Rosser Classification System [26]), and the patient’s follow-up plan. This follow-up plan comprises two sections: first, a mandatory “discharge status” (codified as “continued follow-up in our clinic,” “discharge,” “admission,” or “transfer to another center”), and second, an “elaboration of the discharge status”. Regarding the latter, in case the patient remains being followed-up in our clinic, we can indicate if the patient is going to be referred to our rheumatology nurses (usually for patients with chronic inflammatory diseases treated with immunosuppressive drugs), referred to ophthalmic care (in case of patients treated with antimalarial drugs), or referred to our day-hospital (for administration of iv therapy); in case the patient is discharged, she/he can be referred to primary care, or the rheumatologist can indicate that a scheduled telephone contact to assess the patient’s clinical evolution and response to medication was planned.

In the present study, we have selected patients and discharge episodes based on the following inclusion criteria (Figure 1):Patients with at least one valid discharge from our outpatient clinic, meaning a “discharge status” codified as “discharge” and any “elaboration of the discharge status” that is consistent with that patient´s follow-up plan. Those possibilities include either a referral to primary care, or a scheduled telephone contact. In addition, considering that elaboration of the “discharge status” is not a mandatory variable, meaning that it can be left blank during the contact with the patient, we also considered this situation as compatible with a valid discharge, understood as the physician discharged the patient from our outpatient clinic and did not make any further specifications.Those discharges followed by a time window of at least 12 months.For those discharges followed by an outpatient readmission, there had to be at least one diagnosis shared between the discharge and the readmission visits, meaning the presence of diagnostic ICD9/ICD10 codes associated with the same disease or related group of diseases in both visits. Therefore, the patient did not need to have the same specific code in both visits, but he/she can have two different codes associated with the same disease (list of diseases/groups of diseases can be found in Appendix A Categories, Diagnoses 2nd categorization sheet. More detailed description regarding grouping of diseases can be found in the next section).For those discharges followed by an outpatient readmission, that visit had to take place in the first 12 months after discharge.For those discharges followed by an outpatient readmission, a second cut-off point was established based on the sensitivity and specificity of the mandatory “discharge status” variable to reduce its false positive rate.Finally, because to generate the prediction models the discharge episodes were chronologically split into training, validation, and test datasets, discharges were only included in the latter dataset if the patient had not had a previous discharge in the training or validation datasets; similarly, discharges included in the validation dataset were only included if the patient had not had a previous discharge in the training datasets.

Therefore, in the present study, episodes of discharge between 1 April 2007 and 30 November 2016 were included. The presence of outpatient readmissions was considered until 30 November 2017. 

HCSC Ethics Review Board approval was obtained as a retrospective study and waiver of informed consent was obtained for the use of de-identified clinical records. Furthermore, the study was conducted in accordance with the Declaration of Helsinki.

### 2.2. Variables

Our primary outcome was “outpatient readmission,” defined as a return to our outpatient clinic after a discharge, for at least one common diagnosis, between 2 and 12 months after discharge. All discharges and readmissions fulfilling the inclusion criteria were included in the analysis, not only the first ones.

For the development of the prediction model we considered only those outcomes taking place in the first 12 months after a discharge. By setting this first cut-off point, we likely increased the chance of the patient returning due to either the same episode (because the patient did not experience a complete recovery), or a medium term relapse of the condition that initially motivated the visit to our clinic (because the patient experienced a new episode of the same pathology after a complete recovery). Moreover, because in the long term our intention is to use this model to detect patients more likely to return to the clinic so we can implement some measures after the patient is discharged to prevent readmissions, we considered that we should focus in those episodes taking place in a reasonable period of time after discharge.

To avoid possible “false discharges” (meaning the physician wrongly selected “discharge” as the care plan for the patient but in reality, the patient was scheduled to return), JFU assessed the classification ability of the “discharge status” variable by carrying out a manual review of the clinical notes associated with visits codified as “discharge” or “not discharge,” selected by stratified random sampling. A full description of this method can be found in Appendix A “Sensitivity and specificity of the “discharge status” variable.” We estimated the sensitivity and specificity of the “discharge status” variable considering all visits (regardless the discharge status value) followed-up by a return to the clinic in the following 12 months, and then, we recalculated both metrics excluding the visits associated with a return to the clinic during the first 30 days (1 month), then 60 days (2 months), and so on. Based on the sensitivity and specificity values, we stablished a second cut-off point at 2 months, excluding those discharge episodes followed by a readmission before that time. 

As predictors, we included several demographic and clinical-related variables collected during the patient’s visits using our EHR. Information is stored in a relational database schema composed on several tables including diagnoses and treatments given by the rheumatologists, QoL, administrative and demographic data, visit information, and comorbidity and concomitant treatments prescribed by other physicians outside the rheumatology clinic. Data from those tables were used to create our prediction models.

Regarding QoL, we used the Rosser Classification Index (continuous variable) and its subscale of disability and pain/distress (categorical variables) [26].

Regarding diagnosis given by the rheumatologists, more than eight hundred unique ICD9/ICD10 diagnostic codes were given. These codes were combined and grouped based on diseases, into 64 disease categories (i.e., all codes referred to rheumatoid arthritis were combined into one. This grouping is presented in the Appendix A Categories, Diagnoses 1st categorization sheet).

Regarding drugs prescribed by the rheumatologist, more than five hundred different AEMPS drug codes were prescribed. They were combined according to their active principle into 73 drug categories (this grouping is presented in the Appendix A Categories, Treatments 1st categorization sheet).

Regarding comorbidities and concomitant treatments prescribed by other physicians outside the rheumatology clinic, they are stored as free text. Overall 283 and 132 different comorbidity categories were identified, respectively, and grouped into 195 categories.

Each diagnosis given or drug prescribed at our outpatient clinic were codified as dichotomous variables, representing the presence or absence of each diagnosis or medication at the first visit in the clinic, at the first visit of each episode (defined as the follow-up time between the first visit in our outpatient clinic, a first visit after a previous discharge, and the visit when the patient is discharged), at discharge, and at any visit taking place in the previous 90 or 182 days before discharge (including the discharge visit). For comorbidities and concomitant treatments, each dichotomous variable represented its presence or absence at discharge. Finally, regarding the Rosser Classification Index and its subscales, we used the values at first visit in the clinic, at discharge, and the mean and median values considering all visits in the 90 and 182 days before discharge. 

In order to be included in the analysis, the number of events of a particular diagnosis or drug had to be ≥100 at first visit in the clinic, at the first visit of each episode, at discharge, and at any visit taking place in the previous 90 or 182 days before discharge. In case of a lower frequency in any situation, those categories were re-categorized by affinity into either a new category or added to already pre-existing categories (Appendix A Categories, Diagnoses and Treatments 2nd categorization sheets). Therefore, the number of categories related to diagnosis was reduced to 24, and the number of categories related to treatment was reduced to 16. Regarding comorbidities and concomitant treatments, we considered their prevalence only at discharge, and the number of categories was reduced to 105 and 50, respectively. Overall 403 different variables were included in the models (see Appendix A “Predictors,” and Appendix A Categories: Diagnoses 1st categorization sheet; Diagnoses 2nd categorization sheet; Treatment 1st categorization sheet; Treatment 2nd categorization sheet; Final Comorbidities Categories Sheet; and Variables analyzed sheet).

In Medi-LOG, physicians can only codify the presence of a condition or the prescription of a drug. They cannot explicitly record the absence of a particular diagnosis or medication (i.e., in case of a patient previously seen due to low-back pain, but now suffering only from knee osteoarthritis, in the current visit, the physician will only record the presence of knee osteoarthritis and will not record the presence of low-back pain, as he/she cannot expressly record the absence of a condition). Therefore, in the analysis we assume that the patient only suffers from the conditions explicitly recorded, and when a particular code is absent, we assume that the patient did not present that particular diagnosis or was prescribed with that particular drug. QoL and patient’s follow-up plan are the only mandatory variables and must be completed in each visit by the health professional.

### 2.3. Statistical Analysis

Continuous variables were described using median and interquartile range (IQR), while dichotomous and categorical variables using proportions. Standardized mean difference (SMD) effect size measure was employed to compare the distribution of the predictor variables between discharge episodes with and without outpatient readmission. As it has been suggested, a SMD value smaller than 0.1 indicates a negligible difference [27,28,29].

Models were developed using RF, through R statistical software version 3.3.2 (R Foundation for Statistical Computing, Vienna, Austria) [30] and “randomForest” package version 4.6.12 [31]. This method was chosen because it has shown to outperform other classifiers, it is fairly robust and applicable to big datasets, and requires little modification of parameters prior to modeling [32]. This method generates multiple decision trees based on bootstrap data from the original sample and predicts the outcome of interest based on the majority votes of the individual decision trees. Briefly, the algorithm can be described as follows:Starting from the original data, n bootstrap samples are drawn. Each sample is randomly divided into in-bag data and out-of-bag data (OOB data).From each bootstrap sample, a survival tree is grown using the in-bag data. At each node of the tree, several candidate variables (mtry) are randomly selected, and the candidate variable that maximizes difference between the two daughter nodes is used for splitting the node.The tree is grown until any of the daughter nodes have less that a certain number of events (in our case 1 event).The predicted outcome is determined based on the majority votes of the individual decision trees. Therefore, if more that 50% of the trees predict a particular outcome (in our case outpatient readmission), the forest will predict that the patient will experience an outpatient readmission.

Our cohort was divided into three datasets based on the date of discharge, ensuring at least 200 events (discharge followed by outpatient readmission) in each dataset [33,34]. Those discharges taking place before 30 November 2014, between 30 November 2014, and before 30 November 2015 and between 30 November 2015 and 30 November 2016, were included in the training (containing 69.5% of discharges), validation (14.2%), and test (16.3%) datasets, respectively. To avoid possible bias and over-fitting, each partition only contained information from patients that were not included in other datasets.

Models’ performance was assessed with two types of measures: discrimination, using the area under the receiver operating characteristic curve (AUC-ROC) [35]; and calibration, using calibration curves. Sensitivity, specificity, positive and negative predictive values were also estimated.

In order to identify the model with the greatest discrimination ability and generalizability, we developed in the training dataset several thousand different models following a grid search approach: We chose different pre-defined values of several hyperparameters affecting how the random forest are built, which observations are used, and which predictors are included in the model, and developed a model for each combination of values (Appendix A “Model Fine-tuning,” Figure 2, Appendix A). Overall, up to 61,380 theoretical combinations of tuning parameters were tested. 

All the previously developed models were assessed in the validation dataset. We selected the 10 models with the highest AUC-ROC. To assure the independence of the results and the way the dataset was originally split [36,37], we carried out a ten-fold cross-validation (CV) of those 10 models in the training and validation datasets. Next, those 10 models were assessed in the test dataset, and the model with the highest AUC-ROC in the latter was selected as the final model. Finally, we aimed to reduce the number of variables included in the final model (using the same or similar hyperparameters), in order to generate a reduced final model to facilitate its implementation, following three steps (Appendix A Categories: Reduced Model Sheet):-First, for those variables referred to the same diagnosis given, or drug prescribed by the rheumatologists either at first visit in the clinic, at the first visit of each episode, at discharge, and at any visit taking place in the previous 90 or 182 days before discharge, we chose the one with the highest relative variable importance (VIMP) for each diagnosis and treatment. For those variables referred to disability, distress, Rosser Classification Index, duration of follow-up, and number of visits, the same process was carried out. All comorbidities and concomitant medications were carried out according to the next step.-Second, we considered those comorbidities and concomitant medications referred to the same diagnose and treatments from the first point. We selected either the variable given in our outpatient clinic or the one given by other physicians (i.e., comorbidities) based on the highest relative VIMP. Unrelated comorbidities, concomitant medications, diagnosis given, or drug prescribed by the rheumatologists were carried out according to the next step.-Third and final step, we considered the rest of comorbidities and concomitant medications (those not referred to a diagnosis or treatment from the first point). They were clinically grouped by system or syndrome, and the variable with the highest relative VIMP selected for each system or syndrome. In addition, the variables “number of diagnosis at discharge” and “number of treatments at discharge” were also selected.

This reduced final model was developed using the training and validation datasets combined, and its performance was evaluated in those datasets through 10-fold cross-validation and in the test dataset.

See Appendix A “Methods” for a more detailed description of the statistical analysis. Findings were reported in accordance to the Transparent Reporting of a Multivariable Prediction Model for Individual Prognosis or Diagnosis (TRIPOD) [38].

## 3. Results

### 3.1. Patient Description

In order to estimate the sensitivity and specificity of the “discharge status” variable, 4,096 patient’s discharge episodes were manually reviewed (Figure 3 and Appendix A). Based on how these performance metrics changed when modifying the second cut-off value, we decided to exclude those discharge episodes associated with a readmission occurring in the first two months after discharge. That resulted in a “discharge status” variable (for those episodes associated with a return to the clinic between 2 and 12 months) with a sensitivity and specificity of 0.935 and 0.690, respectively.

Regarding the “discharge status” variable for those episodes not followed by a patient readmission, its sensitivity and specificity were 0.857 and 0.684, respectively.

After applying different inclusion and exclusion criteria (Figure 1), 17,783 patients with 18,662 discharge episodes were finally included in this study. A total of 16,134 discharge episodes were classified as “without outpatient readmission” and 2528 as “with outpatient readmission.” Note that a patient can have discharge episodes allocated in both “without outpatient readmission” and “with outpatient readmission” groups. In fact, 544 have their discharge episodes distributed between the two groups. Table 1 shows the main demographic and clinical characteristics of the discharge episodes. Briefly, discharges with outpatient readmission were more common in women subjects with older age, with greater number of visits and longer follow-up duration, with higher levels of pain/distress and disability, in subject retired or doing housework, in those prescribed with second and third level analgesics, calcium and Vitamin D and gastric protectors, and in those not diagnosed with back pain, or joint pain, or with “no diagnosis.” A more detailed description can be found in the Appendix A.

### 3.2. Model Development, Validation, and Testing

Few differences were observed among the three datasets (Appendix A). Compared with the validation and test datasets, the training dataset had a higher percentage of outpatient readmissions (15% vs. 10% vs. 10%), shorter follow-up until discharge (both in days and in number of visits), lower proportion of active workers (≈55% vs. ≈58% vs. ≈60%) and diagnosis of “chronic polyarthritis” at discharge, 90 and 182 days before (≈3% vs. ≈1% vs. ≈1%), and a higher proportion of treatment with “other DMARDs” (≈2% vs. ≈0.5% vs. ≈0.8%). In addition, comparing with the test dataset, the training dataset presented a higher proportion of treatments with anti-osteoporotic medication and gastric protectors (both at discharge, 90 and 182 days before and at 1st visit), and a lower proportion of diagnosis of “other non-inflammatory diseases” discharge, 90 and 182 days before (≈2% vs. ≈4%).

Based on the combination of different values of a set of hyperparameters, 38,059 models were finally developed in the training and tested in the validation datasets. Appendix A show in a scatter plot all model’s AUC-ROC values in both training and validation datasets without applying and applying the principal component analysis (PCA), respectively. 

Table 2 and Appendix A show the model tuning parameters and performance measures of the 10 models with highest AUC-ROC in the validation dataset. In none of these 10 models Synthetic Minority Over-Sampling Technique (SMOTE) or PCA was applied, and all of them had the same sampsize (50/50).

A slight reduction in the AUC-ROC was observed from the training (with CV) to the test datasets. In addition, our models showed higher specificity but lower sensitivity. The AUC-ROC and calibration curves can be found at Appendix A, respectively. The latter shows that all models tended to underestimate the observed prevalence of outpatient readmissions. 

Regarding the predictors included in these 10 models, the Appendix A shows a heat map with their mean relative VIMP.

Based on the value of AUC-ROC in the test dataset, Model 10 was selected as our final model. It was developed using 250 decision trees, including the same number of cases and controls in each bootstrapped sample used to grow each tree, incorporating all predictors, and randomly selecting 20 variables to split each node of the trees. From training to test datasets the losses in AUC-ROC, sensitivity and specificity were 10.6%, 21.5%, and 2.7%, respectively (Test dataset AUC-ROC = 0.667). Finally, we developed a reduced final model (Figure 4). After applying the mentioned rules, 75 variables remained. Using similar hyperparameter than those of the final model (250 trees, mtry √P, and sampsize 50/50), the developed model showed an AUC-ROC in the training/validation, and test datasets of 0.722 and 0.653, respectively, a performance similar to the final model (Table 2 and Appendix A). Table 3 shows the predictors included in the reduced final model, ranked in order of importance.

## 4. Discussion

We have developed and validated an outpatient readmission predictive model for patients with musculoskeletal and rheumatic diseases followed-up in a rheumatology outpatient clinic from a tertiary center. Based on the predictor classification carried out by Swain et al. [22], our reduced final model included clinical (both chronic inflammatory (such as polyarthritis, rheumatoid arthritis, ankylosing spondylitis or polymyalgia rheumatica) and non-inflammatory diseases (back pain, osteoarthritis, osteoporotic fracture); comorbidities, such as arrhythmias, or use of anti-hypertensive medication), health resources utilization (duration of follow-up, number of visits, number of previous discharges), demographic (age), medication/treatment (being prescribed with antimalarials, oral corticosteroid, other disease modifying anti-rheumatic drugs, antidepressant, 2nd level analgesics, and methotrexate), and patient characteristic (QoL and disability level).

The readmission issue has been thoroughly studied in the inpatient setting, to the point of being considered as an indicator of the quality of care provided; however, in the outpatient setting it remains to be analyzed. From an administrative point on view, the return to the clinic of a patient previously discharged can represent a concern, either due to increased resources consumption or the interruption caused in the continuity of care. However, we recognize that this impact will depend on how the practice or outpatient system is organized. As an example, in outpatient clinics where the system correctly identifies the readmitted patients and either generates an appointment with the discharging physician and/or generates a revision appointment, the administrative impact will be reduced. In addition, it remains to be seen if outpatient readmissions are also associated with a negative impact on the patient’s health, such as extending disability, increasing the chance of disease chronification, the demands to the immediate patient’s support system, and other healthcare resources utilization (including visits to primary, other specialized outpatient clinics, or the emergency care). If that were the case, readmission rate in the outpatient setting can also be used as an indicator of the quality of care provided.

Considering the impact of readmission, the development of tools that are able to predict the individual readmission risk will allow us to identify those patients with higher risk, and can help targeting the delivery of interventions, such as transitional care interventions [39], in order to reduce their chance of readmission. Tree reviews analyzing the performance of prediction models for hospital readmission have been recently published [20,21,22], including more than 100 different models. Kansagara et al. [20] studied 26 validated prediction models developed in an extensive variety of settings and patient populations, most focused on all-cause readmissions, but also on disease-specific readmissions (such as congestive heart failure, acute myocardial infarction, and pneumonia). Swain et al. [22] analyzed 32 models, with the objective of identifying the most important predictors of readmission, and the role, availability, and completeness of EHR structured information for the development of these models. Artetxe et al. [21] included 77 different models, focusing on the data analysis methods and algorithms used.

A wide range of model’s performances have been reported; the accuracy values observed by Kansagara et al. [20] ranged from 0.55 to 0.83, with only six models showing a c-index >0.7. In addition, only nine performed external validation in a new set of patients. Artetxe et al. [21] found 64 models reporting performance metrics, ranging from 0.60 to 0.92 (mean value of 0.71). Only 19% showed value of AUC above 0.75. In our reduced final model, the AUC-ROC in the test sample was 0.653, which can be considered at most a modest discrimination ability. 

Regarding the key predictors of readmission, most models [20,22] used clinical (diagnosis) and treatment data, previous healthcare use, and basic demographic variables. However, few models were developed using variables related to disease severity, health and disability, and social determinants of health. There is a great heterogeneity regarding the predictive capacity of particular variables when different models from different population are compared [22]. However, within the same population, model performance tends to increase with the inclusion of disability measures [40].

In daily practice, our prediction model can be implemented in an interoperable EHR, as it was the main source of data (together with the hospital information system) [41]. The information from the prediction model can be displayed in the same application used by the physician during the outpatient visits, facilitating the visualization of the results and its incorporation in the physician’s decision-making. Our intention is not that this model makes a clinical decision for the rheumatologist, but to focus his/her attention toward those patients that may benefit from a longer follow-up. This model can represent a first step in the prevention of readmission episodes. Further, it can allow us to direct our limited resources to the most vulnerable patients [20]. In fact, some centers have employed successful interventions to reduce inpatient readmissions [42], which can be used as models to prevent readmissions in outpatient settings.

The following are the limitations of the model we developed:(a)Although we included predictors from most categories previously described (such as clinical, demographic, medication/treatment, and QoL) [22], some important predictors, mostly related to social determinants of health, such as income level, household structure, or race, were not included. These variables may impact the patient’s ability to follow the physician recommendations after discharge, increasing the risk of readmission. By not including these variables, two subjects with the same diagnosis, comorbidities, and follow-up duration would have been assigned by our model to the same readmission risk, despite having different income levels, or social support. The absence of social determinants of health in readmission models is common when they are generated using data from EHRs, as they are not usually collected in these tools [43]. Further studies are needed to assess their role in outpatient readmission.(b)Another limitation is that we have included an initial high number of predictors (>400). However, we want to point out that we are also analyzing a high number of discharge episodes (>18,000) and that this is the first analysis of outpatient readmission in the rheumatology setting, therefore there is no evidence regarding which variables should we focus. Finally, we managed to reduce the number of variables used by our reduced final model, which can improve the feasibility of implementation in new setting.(c)Another limitation is the lack of external validation in a different setting, which can limit the generalization of our model. In addition, the same researchers that developed the prediction model carried out the validation analysis, which can lead to bias [44]. However, we want to point out that in our validation and test cohort we included more than 200 events each, the number recommended by some authors to ensure a proper validation [33,34].(d)Class imbalance is an inherent problem when developing readmission prediction models [21]: inpatient readmission is estimated to affect only 20% of admitted patients [9]. Because most of the classifiers assume relatively balanced a priori probabilities for both classes [45], when imbalance is present, the resulting model tends to achieve higher accuracy in the majority class and lower in the minority class [46]. In this study we tried to minimize this issue by oversampling of cases, and by modifying the relationship between the number of cases and controls included in the bootstrapped samples to develop each decision tree.(e)Another limitation is the possibility that patients after discharge instead of being readmitted to our outpatient clinic can be transferred to another center, or to another specialist (such as orthopedic surgery). Therefore, we may be misclassifying patients from the “discharges without outpatient readmission” group. Based on the organization of our regional health system, subjects can request to be attended in the rheumatology department of a different reference area. Approximately 350 subjects from our reference area make this request every year, including subjects never seen in our clinic, previously seen and discharged, and previously seen and lost to follow-up. Assuming that in the previous years (2007–2016) those numbers remained stable, in a worst-case scenario in which all those subjects belonged to the “discharges without outpatient readmission” group and also attended the new centers between 2 and 12 months after discharge, we may be misclassifying up to 21.7% (3500/16,134) of the episodes of this group. Unfortunately, we do not currently have access to the information to identify the precise number of patients we are actually misclassifying. If we were to assume that those patients attending other rheumatology clinics were evenly distributed among the three groups (1166 patients never seen in our clinic, 1166 previously seen and discharged, and 1166 previously seen and lost to follow-up), and of those previously seen and discharged followed the same temporal pattern than those patients returning to our clinic (25% of them retuning in the first 2 months after discharge, 34% returning after 12 month, and 40% between 2 and 12 months), we would be misclassifying only 467 patient of 16,134 (2.9%). We assume the correct percentage could be between both numbers.(f)Because we are using data from routine clinical practice, collected in an environment of heavy workload, errors in codification may exist. To minimize this issue, we included only patients we were confident about that had high quality data, based on data completeness and consistency. Moreover, patient’s care is not formally standardized; meaning that the length of follow-up and the frequency with which patients attend our clinic is based on their clinical manifestations, response, and tolerance to treatment, and the clinical judgment of each physician regarding how the patient is going to evolve. Furthermore, changes regarding the collective management of some musculoskeletal diseases, such as rheumatoid arthritis, can prevent our model from a correct performance if developed in patients managed a certain way and then applied to future patients managed a different way. However, considering that the developed models showed very similar values of AUC-ROC in training, validation, and test datasets, we believe these errors in coding (in case they exist) or the heterogeneity in patient’s management may be systematic and therefore present in all datasets. In addition, changes in patient management do not seem to affect the model’s performance, as we only observed a small reduction in accuracy when comparing between datasets.(g)Another limitation is our definition of outpatient readmission. In order to minimize errors in coding, we excluded those readmissions within the first 2 months after discharge, and therefore we may be discarding early episodes of readmission. However, considering that the median elapsed time between two consecutive visits is close to 2 months (73 days (IQR: 32–133 days)), we can argue that the consequence for a patient that is discharged and returns to the clinic in the first 2 months may be similar to that of a patient not discharged and schedule to return within the first 2 months.(h)In addition, we required the presence of a common diagnosis between the discharge and readmission visits, i.e., the presence of the same or different ICD9/ICD10 codes in both visits but belonging to the same group of diagnoses. However, we did not specifically consider a common diagnosis when despite the readmission being motivated by a process associated or caused by a diagnosis given at discharge, both conditions belonged to different groups.

In addition, we have identified limitations affecting the feasibility of implementing this model in new settings:(a)Musculoskeletal diseases are a very heterogeneous group of conditions. In our outpatient clinic, patients with all sorts of musculoskeletal and rheumatic diseases are attended, including inflammatory and non-inflammatory conditions. The proportion of incident patients with different diagnosis reflects their prevalence in the general population, with most patients being diagnosed with non-inflammatory diseases (such as low back/neck pain, shoulder tendinitis, knee osteoarthritis…). Regarding prevalent patients, the proportion of those with inflammatory diseases increases, due to the care requirements of these conditions (such as rheumatoid arthritis, chronic polyarthritis, polymyalgia rheumatica, spondyloarthropathies, connective tissue diseases…). For the different rheumatology outpatient clinics of the Community of Madrid, a 2011 survey presented in the “Strategic Plan for the Rheumatic and Musculoskeletal Diseases of the Community of Madrid” [47] showed a similar composition regarding the diseases attended in first and review visits. We can hypothesize that for outpatient clinics in other regions of Spain, the composition will be similar. However, there is great heterogeneity regarding clinical practice in these conditions [48]. We can expect this heterogeneity will affect the performance of our model, and therefore, affect its likelihood of being implemented in other settings.(b)Although EHRs represent rich sources of data to develop and improve predictive models, it is essential to consider that our reduced final model still included a large number of variables (>70). However, it is important to point out that all of them are calculated or derived from information routinely collected in our outpatient clinic, and therefore the model’s implementation will not represent an extra effort for the attending physician, as it does not require the collection of new data. Regarding the extension of this model to other rheumatology outpatient settings, there may be issues with the lack of some of the elements collected in our setting. Nevertheless, we have used variables that most EHR in outpatient settings collect: diagnosis, treatments, comorbidities, duration of follow-up, and basic demographic data. The main obstacle will be the absence of a systematic assessment of QoL or disability (in our case we used the Rosser Classification Index) in most EHR. We are aware that the use of this instrument is not as common as others, such as the EuroQol 5D. Regardless, the implementation of our model in a new setting will require an initial manual step of understanding how data is stored, structured, and codified, in order to figure out how to clean, translate, and homogenize the data to be used by the model. However, after this step, besides periodic manual assessments to ensure that this process is carried out with no errors, no further manual work will be required, as these processes can run automatically. Notwithstanding, we recognize this initial manual step and its automatization will limit the application of our model in new settings, as its implementation requires either some abilities and knowledge by the members of the new setting, or external assistance.(c)We have developed a model that identifies patients who are more likely to be readmitted in our outpatient clinic, but we have not shown that those patients have actually a poorer clinical outcome. Unfortunately, we currently have access only to information from those patients who returned to our clinic, and therefore we cannot assess if those coming back have a worse outcome that those who do not, and if our model preferentially identifies this group of patients with poorer outcome, regardless they are readmitted or not. Further studies will be needed to assess this point. Furthermore, considering the influence of the system’s organization in the administrative consequences of outpatient readmission, both circumstances can reduce the impact of our model once it is implemented in a new setting.(d)In addition, multiple factors, such as the definition of the outcome and the population where the model is developed and/or tested are going to play a tremendous influence in model performance. Therefore, trying to use this model in other settings may be associated with a reduction of its discrimination ability. Therefore, a previous testing to assess its performance will be mandatory before implementation in new settings.

## 5. Conclusions

In conclusion, we have developed a prediction model for readmission to a rheumatology outpatient clinic, using data from a departmental EHR. Rheumatologic diseases, such as presence of chronic polyarthritis, osteoporotic fractures, generalized or unspecified osteoarthritis, and hip osteoarthritis were associated with a higher risk of readmission. Conversely, the presence of hand osteoarthritis, neck pain, or muscle disorders were associated with a lower risk. In addition, prescription of treatments such as DMARDs, corticosteroids, 2nd and 3rd level analgesics, or the prescription of a higher number of drugs at discharge were associated with higher risk. Other variables, such a higher duration of follow-up days until discharge since last discharge, homework or retired as occupation, and worse QoL, disability or distress were also associated with higher risk. Finally, comorbidities, such as arrhythmias, column surgery or axial neuropathies were associated with a higher readmission risk.

## Figures and Tables

**Figure 1 jcm-08-01156-f001:**
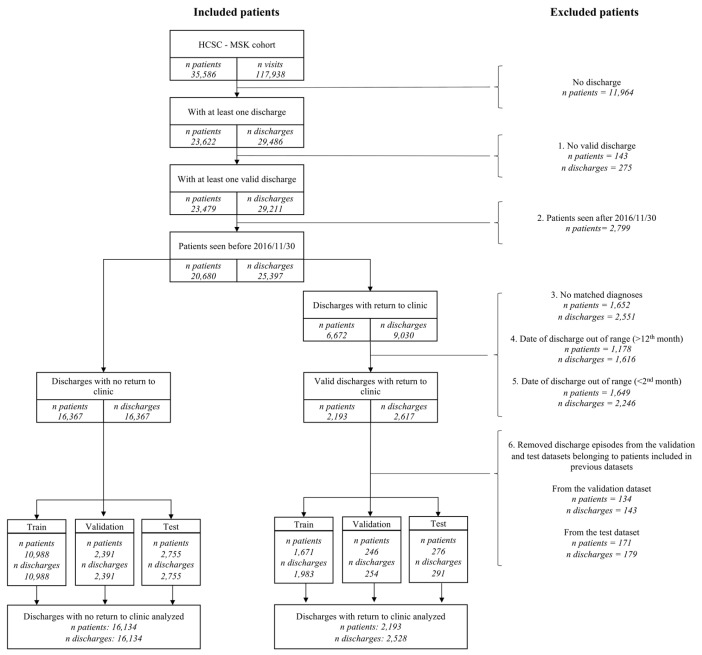
Flow chart of patients included in a study to develop a prediction model of outpatient readmission.

**Figure 2 jcm-08-01156-f002:**
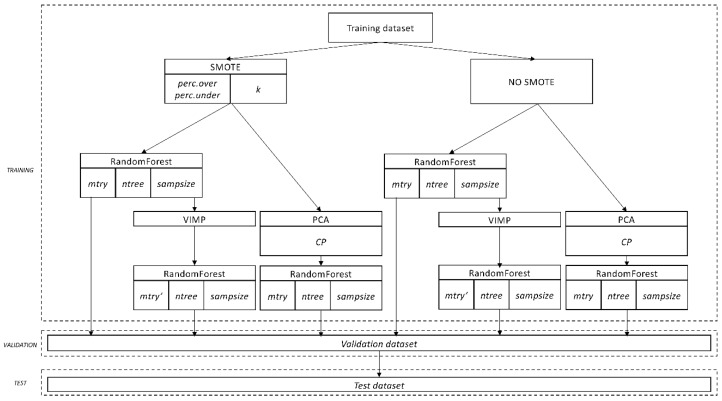
Different tuning parameters and pathways of the developed models. Up to 61,380 different theoretical combinations exist. CP: Cumulative Proportion; PCA: Principal Component Analysis; SMOTE: Synthetic Minority Over-Sampling Technique

**Figure 3 jcm-08-01156-f003:**
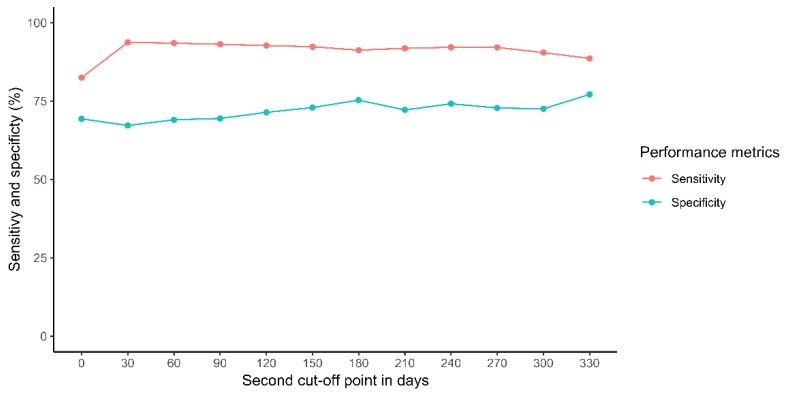
Sensitivity and specificity for the “discharge status” variable based on the second cut-off point (in days).

**Figure 4 jcm-08-01156-f004:**
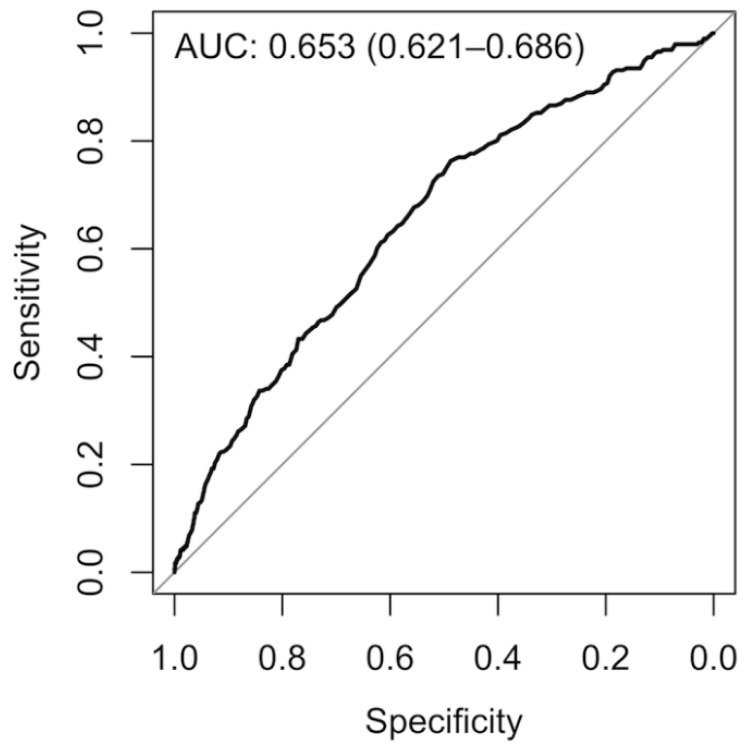
Receiver operating characteristic curve plot of the reduced final model.

**Table 1 jcm-08-01156-t001:** Demographic and clinical-related characteristics at discharge of the patients included in the Hospital Clínico San Carlos musculoskeletal cohort, based on their readmission or not to the clinic. Only the six more prevalent diseases, treatments, and comorbidities were included in this table. Differences expressed as standardized mean differences.

Variables	With Outpatient Readmission *n* = 2528	Without Outpatient Readmission *n* = 16,134	Standardized Mean Difference
Women, *n* (%)	1888 (74.68)	11,266 (69.83)	0.109
Age at discharge, median (IQR)	60.8 (48.8–74.2)	56 (44.3–70)	0.230
Year at discharge, *n* (%)	-	-	0.624
2007	4 (0.16)	738 (4.57)	-
2008	44 (1.74)	1107 (6.86)	-
2009	173 (6.84)	907 (5.62)	-
2010	204 (8.07)	886 (5.49)	-
2011	534 (21.12)	1495 (9.27)	-
2012	510 (20.17)	1768 (10.96)	-
2013	272 (10.76)	2075 (12.86)	-
2014	263 (10.40)	2154 (13.35)	-
2015	262 (10.36)	2471 (15.32)	-
2016	262 (10.36)	2533 (15.7)	-
Number of visits until discharge, since first visit in clinic, median (IQR)	2 (1–4)	1 (1–2)	0.530
Follow-up days until discharge, since first visit in clinic, median (IQR)	42 (0–468)	0 (0–49)	0.334
Follow-up days until discharge, per discharge episode, median (IQR)	0 (0–77)	0 (0–18)	0.223
Number of days elapsed from discharge until readmission, median (IQR)	136 (91–212)	-	-
Rosser Index, median (IQR)	98.6 (98.6–99.5)	98.6 (98.6–99.5)	0.020
Pain/Distress level, *n* (%)	-	-	0.152
None	448 (17.67)	3633 (22.52)	-
Low	1656 (65.58)	10,447 (64.75)	-
Moderate	409 (16.16)	1965 (12.18)	-
High	15 (0.6)	89 (0.55)	-
Disability level, n (%)			0.252
None	911 (36.04)	7714 (47.81)	-
Slight social	1187 (46.95)	5970 (37)	-
Severe social and slight physical	276 (10.92)	1499 (9.29)	-
Moderate decrease in mobility	105 (4.15)	551 (3.42)	-
Severe decrease in mobility	38 (1.50)	292 (1.81)	-
Almost dependent	11 (0.44)	103 (0.64)	-
In bed	-	5 (0.03)	-
Occupation, n (%)	-	-	0.193
Active	1248 (49.34)	9213 (57.10)	-
Housework	721 (28.53)	3744 (23.21)	-
Retired	527 (20.85)	2777 (17.21)	-
Student	32 (1.27)	400 (2.48)	-
Diagnoses, n (%)			-
Back pain	321 (12.70)	2763 (17.13)	0.122
No diagnoses	114 (4.51)	1914 (11.86)	0.211
Osteoarthritis of knee	310 (12.26)	1666 (10.33)	0.063
Pain in joint	144 (5.7)	1876 (11.63)	0.269
Tendinitis lower extremities	247 (9.77)	1193 (7.39)	0.087
Tendinitis upper extremities	478 (18.91)	2622 (16.25)	0.073
Medication use, *n* (%)	-	-	-
Analgesic 1st level	1,019 (40.31)	5969 (37.00)	0.069
Analgesic 2nd and 3rd level	302 (12.02)	1391 (8.62)	0.124
Benzodiazepine	165 (6.53)	1188 (7.36)	0.031
Calcium and vitamin D	350 (13.84)	1002 (6.21)	0.254
Gastric protector	463 (18.31)	2083 (12.91)	0.15
NSAIDs	828 (32.75)	5158 (31.97)	0.019
Comorbidities, *n* (%)	-	-	-
Depression	138 (5.46)	805 (4.99)	0.021
Diabetes mellitus	236 (9.34)	1366 (8.47)	0.031
Dyslipidemia	613 (24.25)	3480 (21.57)	0.064
Hypertension	715 (28.28)	4162 (25.8)	0.056
Hypothyroidism	155 (6.13)	1006 (6.24)	0.004
Obesity	81 (3.20)	634 (3.93)	0.039

NSAIDs: Nonsteroidal anti-inflammatory drugs.

**Table 2 jcm-08-01156-t002:** Area under the receiver operating characteristic curve (AUC-ROC), sensitivity and specificity of the 10 selected outpatient readmission prediction models based on the AUC-ROC in the validation dataset, and of the reduced final model.

Model	Ntree	Mtry	Mtry’	Importance	Nº Pred	AUC-ROC				Sensitivity				Specificity			
						TR	V	CV	TS	TR	V	CV	TS	TR	V	CV	TS
1	100	30	8	1	329	0.744	0.668	0.731	0.662	0.561	0.480	0.535	0.440	0.791	0.743	0.790	0.758
2	100	20	-	ALL	403	0.739	0.668	0.734	0.668	0.536	0.469	0.523	0.416	0.799	0.766	0.801	0.783
3	1000	10	-	ALL	403	0.749	0.669	0.739	0.671	0.554	0.480	0.540	0.471	0.802	0.752	0.794	0.757
4	250	10	-	ALL	403	0.748	0.669	0.737	0.669	0.555	0.484	0.539	0.450	0.798	0.757	0.793	0.755
5	500	10	-	ALL	403	0.748	0.669	0.738	0.670	0.556	0.480	0.541	0.464	0.801	0.759	0.792	0.754
6	100	10	10	20	251	0.736	0.670	0.730	0.663	0.526	0.469	0.533	0.443	0.808	0.782	0.801	0.777
7	100	25	18	1	329	0.739	0.671	0.727	0.665	0.532	0.461	0.511	0.416	0.810	0.775	0.807	0.790
8	500	20	-	ALL	403	0.746	0.672	0.737	0.667	0.530	0.472	0.517	0.412	0.809	0.772	0.805	0.782
9	1000	20	-	ALL	403	0.747	0.672	0.737	0.667	0.532	0.472	0.520	0.416	0.809	0.768	0.805	0.783
10	250	20	-	ALL	403	0.746	0.673	0.736	0.667	0.534	0.472	0.523	0.419	0.806	0.767	0.802	0.784
Reduced final model	250	20	-	-	75	-	-	0.722	0.653	-	-	0.502	0.385	-	-	0.808	0.794

AUC-ROC: area under the receiver operating characteristic curve; CV: cross-validation; Nº Pred: number of predictors; TR: results from the train dataset; CV: cross-validated results from the training and validation datasets; V: validation dataset; TS: test dataset. Sampsize parameter was 50/50 for all models.

**Table 3 jcm-08-01156-t003:** Mean relative variable importance (VIMP) of the predictors included in the reduced final outpatient readmission prediction model.

Category	Predictor	rVIMP (%)	Prevalence/Mean in the “With Outpatient Readmission” Group
Demographic-related	Follow-up days until discharge since last discharge	100	Higher
Treatment-related	(D) Other DMARDs	85.84	Higher
Diagnoses-related	(182) Chronic polyarthritis	58.75	Higher
Treatment-related	(90) Corticosteroid	40.61	Higher
Demographic-related	(B) Occupation	32.66	Higher (house work, retired)
QoL-related	(182) Mean Rosser Index (c)	32.48	Lower
Diagnoses-related	(90) Osteoarthritis of hand	32.32	Lower
Comorbidity-related	(D) Other arrhythmias	31.38	Higher
Treatment-related	(D) Analgesic 2nd and 3rd level	30.02	Higher
Diagnoses-related	(182) Osteoporosis fracture	29.05	Higher
QoL-related	(182) Mean Disability Subscale (c)	28.76	Higher
Comorbidity-related	(D) Column surgery	27.67	Higher
Treatment-related	(D) Number of treatments	27.08	Higher
Diagnoses-related	(182) Generalized or Unspecified Osteoarthritis	25.41	Higher
Demographic-related	(B) Age	24.69	Higher
Comorbidity-related	(D) Axial neuropathy	23.65	Higher
Comorbidity-related	(D) Calcium antagonist	23.56	Higher
Diagnoses-related	(D) Back pain	23.46	Lower
Diagnoses-related	(D) Osteoarthritis of hip	23.24	Higher
Treatment-related	(B) NSAIDs	22.57	Lower
Comorbidity-related	(D) Antidepressant	21.64	Higher
Treatment-related	(D) Gabapentin	21.11	Higher
Comorbidity-related	(D) Hiatal hernia	20.98	Higher
Comorbidity-related	(D) Vitamins	20.84	Higher
Diagnoses-related	(D) Muscle disorders	20.82	Lower
Comorbidity-related	(D) Peripheral nervous system diseases	20.33	Higher
Treatment-related	(B) Gastric protector	20.31	Higher
Comorbidity-related	(D) Antiepileptics	19.39	Higher
Comorbidity-related	(D) Depression	19.11	Higher
QoL-related	(90) Mean Distress Subscale (c)	19.1	Higher
Diagnoses-related	(D) Spondyloarthropathies	18.13	Higher
Diagnoses-related	(D) No diagnoses	17.98	Lower
Treatment-related	(C) Analgesic 1st level	17.59	Higher
Diagnoses-related	(B) Neck pain	17.47	Lower
Treatment-related	(182) Colchicine	16.18	Higher
Comorbidity-related	(D) Other benign tumors	15.96	Higher
Comorbidity-related	(D) Anxiety	15.83	Lower
Comorbidity-related	(D) Calcium	14.46	Higher
Comorbidity-related	(D) Feet diseases	14.46	Higher
Diagnoses-related	(D) Number of diagnoses	14.23	Higher
Comorbidity-related	(D) Bisphosphonates	13.66	Higher
Comorbidity-related	(D) Bronchodilator	13.63	Higher
Comorbidity-related	(D) Cataract	13.42	Higher
Diagnoses-related	(182) Tendinitis upper extremities	12.88	Higher
Diagnoses-related	(182) Tendinitis	12.62	Lower
Comorbidity-related	(D) Kidney failure	12.61	Higher
Comorbidity-related	(D) Hand diseases	12.17	Higher
Diagnoses-related	(C) Other non-inflammatory diseases	11.98	Lower
Comorbidity-related	(D) Lowering uric acid drugs	11.93	Higher
Comorbidity-related	(D) Other endocrine diseases	11.18	Higher
Treatment-related	(182) NSAIDs hard	11.03	Higher
Comorbidity-related	(D) Allergy	10.95	Higher
Comorbidity-related	(D) Other ear diseases	10.74	Higher
Diagnoses-related	(B) Fibromyalgia	10.25	Higher
Diagnoses-related	(B) Pain in joint	10.01	Lower
Comorbidity-related	(D) Benzodiazepine	9.81	Lower
Comorbidity-related	(D) Other spine diseases	9.59	Higher
Comorbidity-related	(D) Knee arthrosis	9.59	Lower
Comorbidity-related	(D) Shoulder diseases	7.94	Higher
Comorbidity-related	(D) Urological surgery	7.36	Higher
Comorbidity-related	(D) Other virus infection	7.32	Higher
Comorbidity-related	(D) Symptomatic slow action drugs for osteoarthritis	6.78	Lower
Comorbidity-related	(D) Pregnancy	3.34	Lower
Comorbidity-related	(D) Other psychiatric conditions	1.76	Higher
Diagnoses-related	(182) Other connective tissue inflammatory diseases	1.72	Higher
Diagnoses-related	(D) Tendinitis lower extremities	1.47	Higher
Comorbidity-related	(D) Iron	1.12	Higher
Comorbidity-related	(D) Smoking habit	0.89	Lower
Diagnoses-related	(C) Crystal arthropathy	0.15	Higher
Demographic-related	Sex	−0.09	Higher
Diagnoses-related	(C) Gout	−0.46	Higher
Comorbidity-related	(D) Other dermatological diseases	−1.95	Higher
Diagnoses-related	(90) Osteoarthritis of first carpometacarpal joints	−2.68	Lower
Comorbidity-related	(D) Family history	−3.41	Higher
Diagnoses-related	(182) Osteoporosis	−3.62	Higher

(B): presence at baseline; (C): presence at the first visit of the episode; (c): mean/median value considering the discharge observation; (D): presence at discharge; (90): presence of or mean/median value of all the observations registered in the last 90 day before discharge; (182): presence of or mean/median value of all the observations registered in the last 182 days before discharge; DMARDs: disease-modifying antirheumatic drugs; NSAIDs: nonsteroidal anti-inflammatory drugs; QoL: quality of life; rVIMP: relative variable importance.

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
