# Peer review of "Outpatient Readmission in Rheumatology: A Machine Learning Predictive Model of Patient’s Return to the Clinic"

_jcm, 2019, doi:10.3390/jcm8081156_

Round 1

Reviewer 1 Report

The authors have adequately addressed my comments and concerns

Reviewer 2 Report

I thank the authors for trying to address my concerns, but I am not satisfied that all of them have been attenuated.

I remain unconvinced about the methods used and do not believe that the authors have made the paper sufficiently more easy to digest. My specific comments made in response to the rebuttal letter are detailed below:

Specific comments 

1-7, 9-14, 16, 18, 20-23, 25, 27-28, 37-38, 40-41: No further comments.

8: I am afraid this justification is not sufficient. Unless the data are already peer reviewed and available it remains hypothetical. You have mentioned above that other tools have sought to predict readmission and they must have justifications within them about why it is important. You need to better justify this. If this paper is as a result of other work, you really should publish the other work first. It is not acceptable to be unclear for this reason alone.

15: I think you should be more upfront about the fact that data collection closed in December 2017. This is an important reason for the lack of data after this date and inclusion of this would have prevented my query in the first place.

16: I think the modifications you have made are still confusing. this suggests you can have a valid discharge if you are coded as "discharge" AND:

a) have a referral to primary care, OR

b) have a telephone call arranged, OR

c) have no specification.

How can no specification be a valid discharge? I also think the term "compatible elaboration" is unhelpful and unclear.

19: You have not answered my question. I understand that common diagnosis means shared between two visits. What I asked is how you have determined this. It is possible that someone comes in with diagnosis A, but then is readmitted for diagnosis B, however they can only have diagnosis B because they have diagnosis A.... how is this classified within your "common/shared" diagnosis?

24: How have you come to the decision that readmission after 12 months is not clinically relevant!? It is not immediately clear that JFU are author initials as you have written "we".

26: I appreciate the inclusion of the predictor section, but I think it could be improved. What do you mean by "more than eight hundred unique ICD9/10 diagnostic codes have been given SO FAR" - have you not finished data extraction? Why have you selected >100 discharges. This figure feels suspiciously low given the sample size. I remain unconvinced that 457 variables is an appropriate number.

29: This has not addressed my point. I am not concerned about duplicate people, rather I am concerned that any temporal affects of changes to treatment paradigms may unduly affect your algorithm. If you create something in groups who had no biologics and test them in groups with biologics, I do not see how you are testing in a comparable cohort.

30: I do not think this is any more digestible.

31: I do not understand your rebuttal. What is a "visit not classified as a discharge"? I do not think it is acceptable to justify decisions on work that is unpublished.

32: Though this gives me a BIT more information I cannot know the direction of the associations without looking at the supplementary documents. I should be able to know from this paragraph whether, for example, those with readmission are more likely to be older women. 

33/34:  The description you have provided is fine, but it does not tell me what the model contains. It seems to me if your final best model cannot be listed in the manuscript it really does have too many variables.

35: It doesn't really matter if the data are collected in EHRs often, the data still need to be cleaned, sometimes coded and appropriately structured to run your algorithm. This makes it unlikely that your model will be used in the future and that completely limits its use. Who are you proposing would have the time to run this and what would they do with the output.

36: Point 8 does not address this issue. You have not indicated whether your model is able to identify those who have poorer outcomes.

39: By what means have you decided it is unlikely that patients belong to that group?! 21.7% is a high proportion of people and this should not be so readily dismissed.

42: I think you should do better than just saying you dont think things are a problem.

43: I remain unconvinced that you can single out any of these predictors when you have >300.

Reviewer 3 Report

The authors have done well in addressing my original comments. I have only one minor comment.

I am still unclear where the 17,772 patients reported in the Results section is coming from.

The authors report: "After applying different inclusion and exclusion criteria (Figure 1), 17,772 patients with 18,648 discharge episodes were finally included in this study."

Looking at the revised flow-chart, the final numbers are 16,135 patients with 16,135 discharge events for those that had no return, and 2,181 patients with 2,513 discharges for those that did return. That totals 18,316 patients with 18,648 discharges. The reported discharge number is the same, but can the authors check whether the quoted 17,772 is correct.

Author Response

This manuscript is a resubmission of an earlier submission. The following is a list of the peer review reports and author responses from that submission.

Round 1

Reviewer 1 Report

I read the article with great interest. This is an interesting attempt to assess readmission. I believe that the presented model can be a useful tool in preventing episodes of readmission in everyday practice and, what is very important, become an inspiration to create new, even better models.

Reviewer 2 Report

General comments:

This paper is overly complicated and the purpose of it is not clear. The authors imply (and directly state) that the impact of outpatient readmission is not known, so it is not clear to me why they are developing a predictive model to identify people at risk of readmission.

There is much ambiguity in the data selected as predictors and at no point in the main manuscript is there a complete discussion of all variables included in the model. Instead the most important variables are included in Table 3. There are 45 predictors in this table and we have no idea how many in the full model. It is not clear whether it is reasonable to have such a big model and what the utility of it will be given the amount of data needed.

A number of assumptions regarding data cut offs and presence of data feel fundamentally flawed. The method and the meaning  behind the procedures undertaken are poorly communicated to a clinical audience and one is left feeling that the machine learning method was selected more because it is “fancy” or “complex” than because it is necessary to answer the research question. Why would a basic prediction model not suffice?

There are grammatical errors and spelling mistakes throughout and the manuscript would benfit from a thorough proof read.

Specific comments:

Abstract – page 1 line 20: I wonder if it would be more clear to say “based on the Random Forest algorithm”, rather than just “based on Random Forest”

page 1 line 21-22: is the “(outpatient readmission)” necessary? I’m not sure it clarifies anything.

Page 1 line 22-23: I find this sentence a bit clunky. Would it be better to move “and followed-up until November 2017” to the end of the sentence? Can you provide exact dates?

Page 1 line 24: Why only 2-12 months?

Page 1 line 24-25: How were they split? – randomly, consecutively etc.

Introduction – page 2 line 46: The use of “was reported” suggests you are talking about one specific example, I think you want to say “has been reported”

page 2 line 47-49: it is not clear to me what the value-base healthcare approaches are that you refer to. I also don’t really understand how paragraph one links to paragraph two. In this first section you are talking about readmissions from inpatient procedures, but then go on to talk about increases in outpatient visits… are you suggesting that the value-based healthcare approaches now require people to present as outpatients following their stay as an inpatient?

Page 2 line 50-56: is this all a hypothetical impact? You say it “could be” associated with… but unless it is true that premature discharges have negative effects I am not sure this paper has merit.

Page 2 line 58-59: I would reorder this and mention that the end goal is the targeted delivery of interventions to prevent outpatient readmissions in those at greatest risk, but in order to do that you need first to know who is at risk.

Page 2 line 64-65: I am not familiar with the term “day-to-day clinical practice cohort” is there a more traditional way to describe this study population?

Page 2 lines 6-70: I think this section feels cumbersome. Is it not just sufficient to say “The Hospital Clínico San Carlos musculoskeletal cohort (HCSC-MSKC) is a day-to-day clinical practice cohort that includes all subjects seen at the rheumatology outpatient clinic of our centre since [IMPLEMENTATION DATE].” Can you provide the actual date of implementation?

Please also be more transparent about what you mean when you say “due to data completeness”.

Please also give examples of what you mean by structured and unstructured data

Page 2 lines 70-79: This section is also a bit clunky and is one really long sentence. The use of codified could be replaced with comprises – i.e. “this plan comprises two sections”.

                Suggest rephrasing into something like:

“The patient’s follow-up plan comprises two sections: first a mandatory discharge status (continued follow-up in our clinic, discharge, admit, or transfer) and second an elaboration of discharge status….” Then in a second sentence you can discuss the options given in the second section.  

Page 2 line 80: why was this timeframe selected? The final inclusion date is around 2.5 years ago, so it seems there would be an opportunity to have collected more data?

Page 2 line 81: Do you just mean “Data regarding outpatient readmissions were obtained until November 30th 2017”? Why only until this date? Given that this is 18 months ago it seems like you may be missing out on lots of potential readmissions.

Page 2 line 87, also figure 1: what do you mean by “valid discharge”? Later in figure one you talk about discharges and valid discharges and the difference is unclear.

Page 2 line 90: Bullet point 2 seems redundant as you have already mentioned this above.

Page 2 line 91-92: how did you determine common diagnoses? What if the readmission was for something associated with an original diagnosis, but not the original diagnosis itself?

Page 2 line 93-97: It is not clear to me what bullet point 4 means. Why did it have to occur within 12 months? Can you better explain why the second discharge cut-point was needed? I think these should be listed as separate points.

Page 3 line 99-102: I am not clear about what the point you are trying to make here is.

Figure 1 – I do not think this diagram is self-explanatory, particularly at the lowest two levels.

Page 3 line 111-112 - I think it is more accurate to specify that your primary outcome is defined as a return to outpatient clinic for at least one common diagnosis, between 2 and 12 months after discharge. If after readmission someone was again discharged and then readmitted do they contribute twice? Or is it readmission after first discharge?  

Page 3 line 113-125 – I find this section hard to follow. It feels counter intuitive to say that you want to increase the chance of capturing short/medium relapses, but also to have excluded some of the time in which they may have relapsed in the short term. Is the only reason that readmission within the first two months could happen due to the incorrect selection of “discharge”? How many discharges were excluded for this reason and how likely is it that none of them were due to relapse?

Page 4 line 123-124 – you use “sensibility” frequently here and throughout the supplementary documents, but I think you mean sensitivity. I do not think that the quoted supplementary file helps to clarify the procedure.

Page 4 lines 126-130 – I don’t think that the description of predictors should be relegated to supplementary files. I think this is essential information.

Page 4 line 131-134 – The sentence “In Medi-LOG, variables are recorded only when present.” Does not make sense, do you mean that you could only extract data if it had been recorded by the physician? How reasonable is it to assume that the patient does not have the condition or medication due to missingness? This is concerning given that you excluded follow-up time in case of physician error!

Paged 4 line 136-137 – this is not helpful given that you have not been upfront about the variables in the above

Page 4 line 145 – how appropriate is splitting by time rather than person? The inception date of this cohort predates biologics and treatment paradigm changes such as this may have a substantial impact on rates of relapse. Also splitting by time may mean that things like clinic staff and their biases have an impact on the performance of the algorithm. If the models don’t perform well in thr test and validate datasets how sure can we be that this is not just due to the impact of temporal differnces?

Page 4 line 152-186 – I am not sure this section will be easily digestible for many of the journal’s readers. Please consider giving a less technical overview explaining why you have done what you did.

Figure 3  - the benefit of selecting exclusion of the first two months versus the first month seems negligible when looking at the values.

Results – this entire section is quite hard to follow. Table 1 includes a lot of data that you don’t discuss, yet you do discuss differences between the datasets without presenting the data.

Page 10 line 240-243. Does Table 3 show “the most important predictors” or all predictors ranked in order of importance?

Throughout - At no point do you seem to discribe the final model..?

Throughout - What use is a model with so many predictors?

Discussion – with the exception of readmission, it is not clear whether your model does actually identify people at risk of poor outcomes. That is, you do identify people who are readmitted, but it is still unclear whether those people have poorer outcomes than people who are not. This is particularly important for the section about the impact of your model.

Page 12 Line 297 – the first limitation is a limitation of the research field, not your work which seems disingenuous. It is also unclear to me why you have created the predictive model if the impact of readmission in outpatients is not known.

Page 12 line 301 – it does not feel sufficient to say “some important predictors” were not included. It would be more reasonable to be specific about the variables and why they are important. If the variables are so important, was the EHR an appropriate data set to conduct this analysis in?

Page 13 line 316-319 – surely this is a fundamental problem.

Page 13 line 321-324 – how can you be confident of this? If the errors are systematic it would not be correct to say they have minimal importance, just that they are present in all datasets

Page 13 line 325-327 – I think this is a fundamental problem.

Limitations – in general this section is problematic. Listing the limitations without justification or adequate discussion is not reassuring. Many of these are fundamental to the interpretation aand utility of the model.

Conclusions – The conclusion of this work is disingenuous. Singling out QoL (predictor 23 of 45) as being of particular importance is spurious. Also in the methods you have used Rosser Classification Index to represent health status, not QoL. These are not identical constructs.

Reviewer 3 Report

I enjoyed reading this manuscript that investigated the use of machine learning methods, specifically random forests, in order to predict outpatient readmission rates at a Rheumatology clinic using electronic health records of patients recruited to a clinical practice cohort (HCSC-MSKC).

I found the introduction to be lacking in detail with respect to what other modelling of readmission rates had been performed in the medical literature, particularly with respect to the rheumatology setting. I do note that paragraphs 2,3 and 4 of the discussion does include detail on other studies looking at modelling predictors of readmission, and wonder if they would be better suited for the introduction?

Could the authors please clarify the numbers quoted in the flow diagram (Figure 1.) and those quoted in the results section regarding the final analysis numbers (17,772 patients with 18,648 discharges).

With respect to the statistical methodology, I was wondering if the authors could clarify the decision to split the data into test, validation and training datasets based on date of discharge, rather than splitting the data randomly? I note that Supplementary Table S10, which compares the different data sets, reports higher percentage of outpatient readmissions, which could be explained by the fact that earlier dates of discharge were used, resulting in longer follow-up for these patients. Unless the methodology can be sufficiently justified, I would be interested to see if randomly splitting the data could minimise the potential differences between the training and validation datasets, which could lead to smaller errors in the development of the prediction models.

Furthermore, can the authors explain the use of cross-validation on the training data only, and not on the training and validation data combined? This method would enable the k-1 data to act as the validation dataset, so it would make sense to incorporate the validation data when performing cross validation.

With respect to the results section, can the authors provide further clarity on the decision to include only those readmissions that occurred after 2-months (60-days)? In the discussion, a limitation listed is that your definition of outpatient readmission is restricted to those occurring after 2-months and therefore you may discard any readmissions occurring earlier on. However, looking at the sensitivity and specificity plot in Figure 3, it would appear to me that there is minimal change in sensitivity and specificity when using 30-days as the cut-point. Could this not be used instead given the limitation described when using 60-days as the cut-point?

Other minor points would be the need for references in the introduction to evidence claims made (e.g. ‘Premature outpatient discharges could have negative impacts at multiple levels, extending disability and increasing the chance of disease chronification, the demands of the immediate patient’s support system and healthcare resources utilization.”), and general proof reading to check for English grammar and spelling.